# The Impact of Transport Infrastructure on Rural Industrial Integration: Spatial Spillover Effects and Spatio-Temporal Heterogeneity

**Han Zhang** * and **Dongli Wu**

College of Economics and Management, Shenyang Agricultural University, Shenyang 110866, China; wdl@syau.edu.cn
* Correspondence: zhangh@stu.syau.edu.cn

**Abstract:** Industry convergence is the future trend of industrial development in rural areas and is conducive to high-quality agriculture development. To explore the development dynamics of industry convergence. This paper selects data from 31 provincial administrative regions in China from 2009 to 2019. It uses the entropy power method to measure the development quality of rural industrial integration in China and empirically studies the impact of transportation infrastructure on rural industrial integration using a spatial panel autoregressive model. The study found that: (1) from 2009–2019, the development quality of rural industrial integration is on the rise, but the development is uneven between regions; (2) transport infrastructure strongly promotes the development of rural industrial integration; (3) with the help of transport infrastructure, rural industrial integration in this region will improve the quality of rural industrial integration in the surrounding areas; and (4) the impact of transportation facilities varies in different regions and at different stages of development of rural industrial integration. The results of this paper are beneficial to improving transportation infrastructure planning and exploring the driving force of high-quality agriculture development, enriching the research of spatial land use, and providing valuable insights for developing industry convergence in other countries and regions.

**Keywords:** transport infrastructure; rural industrial integration; spatial spillover effects; spatio-temporal heterogeneity





## 1. Introduction

In 1963, Rosenberg introduced the concept of industrial convergence [1]. In China, industry convergence is applied in rural areas and is called the integrated development of the primary, secondary, and tertiary industries in rural areas (in this article, it is referred to as rural industrial integration). As a new development model, rural industrial integration operates through integrating resources and applying new technologies to improve agricultural production efficiency and develop secondary and tertiary industries in rural areas. Rural industrial integration is an important initiative in China to promote the industrial development of rural areas, achieve high-quality agricultural development, and efficient use of land. According to the data released by the National Bureau of Statistics of China, China is a traditional agricultural country with a rural population of about 564 million. This paper collates the reports and documents published by the Chinese government network on rural industrial integration. To promote the development of modern agriculture in the context of a large country with small farmers, the 2014 China Central Rural Work Conference proposed "introducing modern industrial organization methods, such as industrial chains and value chains, into agriculture and promoting the integrated development of the primary, secondary, and tertiary industries in rural areas". In 2015, the General Office of the State Council promulgated the *Guidelines on Promoting the Integration of Rural Primary, Secondary, and Tertiary Industries*. Since then, all versions of the *No. 1 Central Document* issued by China

from 2016 to 2022 have mentioned promoting and improving the development of rural industrial integration. How to promote the development of rural industrial integration has received extensive attention from the Chinese government and academia.

Since rural industrial integration was proposed, the academic community has also been searching for the driving force for its development [2]. The integration of rural industries began with technology integration between industries, with technological innovation as its driving force [3]. As a new driving force for industrial development, industrial convergence realizes the linkage development among industries through the agglomeration of different industries and the deployment of elements [4]. Based on the perspective of urban–rural interaction, Li believes that the driving forces of rural industrial integration are location and resource advantages and equalization of basic public services, etc. [5]. As a development model, industrial integration involves factors such as the flow of factors and the optimization of industrial structure [6], while the growth of agriculture in different regions needs to be based on the location and resources of the region [7]. However, there is little evidence in the literature on how transport infrastructure affects the development of rural industrial integration in various regions, and the lack of theoretical research has limited the development of rural industrial integration.

Also, literature on promoting the effects of transportation infrastructure on industrial and economic development is summarized. Transport infrastructure breaks regional boundaries by reducing transport costs and influencing regions' development [8]. As research progresses, some scholars propose that transport infrastructure promotes the flow and allocation of capital [9], labor [10], technology [11], and other factors, which, in turn, have a driving effect on regional economic growth [12] and industrial development [13]. The impact of transport infrastructure on regional economic growth occurs not only in urban areas [14] but also in rural areas [15,16]. The impact of transport facilities on regional industrial development is not only on industrial productivity [17,18] and the total factor productivity of the service industry [19], but transport facilities also promote the gross agricultural product [7]. However, some scholars have put forward a different viewpoint, arguing that the driving effect of transport infrastructure on economic and industrial development does not have a long-term effect [20]. Based on a study of the demand for transport infrastructure in different regions, Fogel concluded that there was no greater driving effect of transport infrastructure on regional development [21]. It is even possible that transport infrastructure has a negative impact on regional development [22].

Does the improvement of transport facilities have an impact on the development of rural industrial integration? What impact will it have on the development of rural industrial integration? The answers to these questions are of great practical significance in promoting rural industrial integration and high-quality agricultural development. This paper selects data from 31 provincial administrative regions in China (excluding Hong Kong, Macao, and Taiwan) from 2009 to 2019. It uses the entropy weight method and spatial panel autoregressive model to empirically test the impact of transport infrastructure on the integration of rural industries. The findings of this paper can provide some policy suggestions for improving transportation planning, achieving rural industrial integration, high-quality agricultural development, and scientific use of land.

Based on the above analysis, the marginal contributions of this paper are threefold: firstly, the impact of transport infrastructure on rural industrial integration is empirically tested by placing transport infrastructure and the development quality of rural industrial integration in a model; secondly, considering the possible heterogeneity of the impact of transport infrastructure on rural industrial integration, an attempt is made to divide the sample according to different regions and discuss the impact of the degree of perfection of transport infrastructure on the development quality of rural industrial integration separately. Moreover, the development of rural industrial integration is divided into different stages using the time of policy document release as a node. The impact of transport infrastructure on rural industrial integration at different time stages is further analyzed; and, thirdly, to address the problems caused by spatially lagged terms, an attempt is made

to analyze the impact of transport infrastructure on the integration of rural industries and the differences in the impact of different temporal and spatial dimensions through a spatial panel autoregressive model.

## 2. Research Hypothesis

Traditional economic theories focus on the accumulation of factors and ignore the importance of factor flow to economic and industrial development, making it difficult to serve as a theoretical basis for verifying the impact of transport infrastructure on the integration of rural industries. At the same time, the integration of rural industries is driven by technological innovation, through the allocation of factors and industrial restructuring, to strengthen inter-industrial links and, thus, form a model of integrated development between industries in rural areas. As a development model, the development of rural industrial integration is also influenced by location and infrastructure, so the efficiency of factor flow will affect the quality of development of rural industrial integration. In the integration of rural industries, various factors of production, such as capital, labor, and technology, will flow between regions. Transport infrastructure can reduce the transaction costs of factors between regions, break through the limitations of time and space distance [23], and influence the development of rural industrial integration. At the same time, improving transport infrastructure can help break down market segmentation and expand the scale of regional markets [24], thus further promoting the development of rural industrial integration. The influence path is shown in Figure 1. This view also leads us to Hypothesis 1:

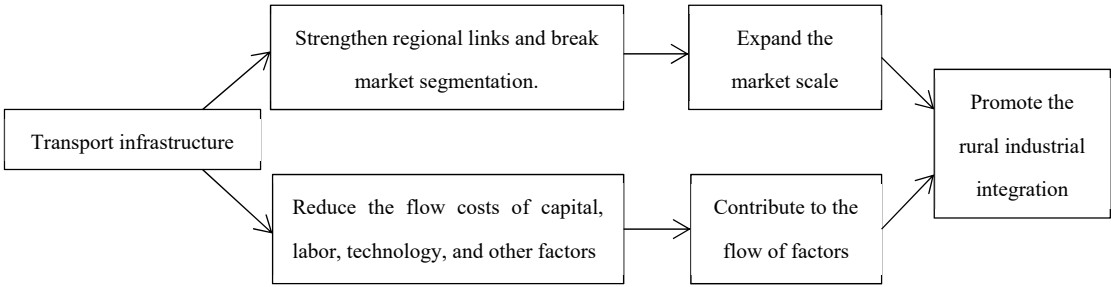

**Figure 1.** Influence Path.

**Hypothesis 1 (H1).** *Transport infrastructure has a positive impact on the development of rural industrial integration in the region.*

Transportation facilities can reduce transaction costs and direct the flow of factors to the most efficient areas and then form clusters in areas with a high level of rural industrial integration development and in areas with a low level of rural industrial integration development. The development of factor inputs and industries between regions is not independent of each other but is spatially dependent. To study the impact of transport infrastructure on the integration of rural industries, we need to pay attention to its spatial externalities. On the one hand, transport infrastructure is conducive to the flow of factors, while road and railway transport, as they pass through several regions, can, therefore, connect different regions so that the regions are linked through transport facilities and strengthen the inter-regional exchanges [25]. Thus, regions with a high quality of rural industrial integration development will, through the diffusion effect, be able to drive the surrounding regions with a low quality of development. On the other hand, however, attention needs to be paid to the existence of the siphoning effect [26]. Through transport infrastructure, regions with a high quality of rural industrial integration development take advantage of their resources and location to attract factors from surrounding regions to cluster in their regions. They are creating a negative spatial externality on the development of rural industrial integration in surrounding regions—transport infrastructure causes

this adverse effect. However, considering the different economic scales and differences in market size among regions, transport infrastructure is conducive to breaking the interregional market segmentation and expanding regional market size [27]. There is a positive spatial externality impact which will also benefit the development of rural industrial integration in the surrounding areas. We, therefore, propose Hypothesis 2:

**Hypothesis 2 (H2).** *There is a positive spatial spillover effect of transport infrastructure on the development of rural industrial integration.*

## 3. Research Methodology

### 3.1. Econometric Model

Drawing on Boarnet's study [28], this paper constructs a model for studying the quality of rural industrial integration development based on the Cobb–Douglas production function and considers the factor of transportation infrastructure, assuming that region is independent and individual and that the quality of rural industrial integration development in the region is determined by the situation of capital, land, technology, and other factors in the region; at which point we obtain Equation (1):

$$Y = A^{\beta} L^{\alpha_1} K^{\alpha_2} \delta(X) \tag{1}$$

In Equation (1), $Y$ indicates the quality of development of rural industrial integration, $A$ denotes the level of technology, $L$ denotes labor input, $K$ denotes capital input, $\delta$ is a random disturbance term, and $X$ denotes the construction of transport infrastructure.

By taking the logarithm of the previous equation, Equation (1), and introducing transport infrastructure as a variable, we obtain Equation (2):

$$lnY = \beta lnA + \alpha_1 lnL + \alpha_2 lnK + \delta lnX + \varepsilon \tag{2}$$

In Equation (2), $\delta$ is the variable $X$ coefficient, $\varepsilon$ is the random perturbation term, and other variables mean the same as in the previous Equation (1). We also need to consider the problem of spatial externality [29,30], so we need to introduce the spatial lag term of the dependent variable to obtain Equation (3):

$$lnY = \rho W lnY + \beta lnA + \alpha_1 lnL + \alpha_2 lnK + \delta lnX + \varepsilon \tag{3}$$

In Equation (3), $\rho W lnY$ denotes the dependent variable spatial lag term; $A$, $L$, and $K$ denote technology level, labor, and capital, which are control variables; $X$ denotes transport infrastructure, which is the core explanatory variable; and $\varepsilon$ is a random disturbance term.

Based on the previous literature review and theoretical analysis, we establish an econometric model to analyze the impact of transport infrastructure on the quality of rural industrial integration development, controlling for other factors. We assume that the development of rural industrial integration in region $i$ is influenced by the transport infrastructure in the region. At the same time, with the help of transport facilities, the rural industrial integration in the region is also influenced by the rural industrial integration in the surrounding areas. There is a spatial spillover effect [30]. Based on Equation (3) above, and, since taking logarithms does not affect the trends and interrelationships of variables, we establish the following econometric model:

$$Y = \mu + \gamma WY + \varphi X + \theta H + \varepsilon \tag{4}$$

In Equation (4), $Y$ denotes the development quality of rural industrial integration; $X$ denotes transport infrastructure, which is the core explanatory variable of this paper; $W$ is the spatial weight matrix; $H$ is the control variable, mainly capital, technology level, labor, etc.; and $\varepsilon$ is the residual disturbance term.

*3.2. Measurement Method*

Based on data from 31 provincial administrative regions in China from 2009 to 2019, this paper empirically examines the impact of transport infrastructure on rural industrial integration with the help of a spatial panel autoregressive model. To this end, this paper constructs an evaluation index system for rural industrial integration development quality. It measures the quality of development of rural industrial integration in China through the entropy weight method [31]. As the units and magnitudes of the observations of each indicator in the index system are different, to facilitate comparison and weighting, it is necessary to do the normalization process and transform the data of each secondary indicator into dimensionless values [32]. The specific steps are as follows.

Firstly, according to the relationship between secondary indicators and the integration of rural industries, they are classified into positive indicators and negative indicators. Positive indicators are those with higher values, representing better development of rural industrial integration, such as the per capita gross product of people working in agriculture, forestry, animal husbandry, and fishery, and the per capita disposable income of farmers households. Negative indicators are those with higher values, representing the lower quality of integrated development of rural industries, such as the amount of fertilizer used per hectare and the proportion of the sown area of food crops to the total sown area. Specific indicators are standardized as follows.

Positive indicators:

$$U_{ij} = \frac{Z_{ij} - min\{Z_{ij}\}}{max\{Z_{ij}\} - min\{Z_{ij}\}} \qquad (5)$$

Negative indicators:

$$U_{ij} = \frac{max\{Z_{ij}\} - Z_{ij}}{max\{Z_{ij}\} - min\{Z_{ij}\}} \qquad (6)$$

In Equations (5) and (6), $U_{ij}$ denotes the value of the jth indicator in the ith region, after normalization, obviously, $0 < U_{ij} < 1$, dimensionless, where max $\{Z_{ij}\}$ and min $\{Z_{ij}\}$ denote the maximum and minimum values of indicator *i* in all samples, respectively.

Secondly, based on the standardized data, the entropy weight method was applied to determine each indicator's weights and derive the scores of each indicator. The formula is as follows:

$$W_j = \frac{1 - C_j}{\sum_{j=1}^{n}(1 - C_j)} \qquad (7)$$

In Equation (7), $C_j = -N\sum_{i=1}^{m}(P_{ij} \times lnP_{ij})$, $N = \frac{1}{lnm}$, and $P_{ij} = \frac{U_{ij}}{\sum_{i=1}^{m}U_{ij}}$, *m* is the number of regions.

Finally, based on determining the weights of each indicator, the overall score of rural industrial integration in region *i* was obtained for each year.

$$Score_i = \sum_{j=1}^{n}(W_j \times U_{ij}) \qquad (8)$$

In Equation (8), *n* is the number of indicators.

*3.3. Evaluation Index System for the Development Quality of Rural Industrial Integration*

There is not yet a unified index system for measuring rural industrial integration development quality and level in academia. The main methods for measuring the integration of rural industries are the Herfindahl index method and the entropy weight method [31,33]. In 2015, the General Office of the State Council of China promulgated the *Guidelines on Promoting the Integration of Rural Primary, Secondary and Tertiary Industries*, which provides an overall plan for the integration of rural industries in six aspects:

- improvement of the general development level;
- completeness of industrial chains;
- diversity of functions;

- the richness of business modes;
- close linkage of interests; and
- integration of industries and cities.

At the same time, through academic research on industrial integration, technological innovation is considered its driving force. Based on the above-mentioned study's methods and theories, we choose the entropy weight method to establish an indicator system based on five aspects:

- intra-agricultural integration;
- extension of the agricultural industry chain;
- expansion of agricultural multifunctionality;
- industrial technology penetration; and
- interest linkage mechanism.

Also, we draw on relevant studies [34,35] and identify secondary indicators. Table 1 demonstrates the specific indicator system.

**Table 1.** Evaluation index system for the quality of development of rural industrial integration.

| Tier 1 Indicators | Secondary Indicators | Unit |
|---|---|---|
| Intra-agricultural integration | Machine cultivation area/Total sown area of crops | % |
| | Total output value of agriculture, forestry, animal husbandry, and fishery/Number of people employed in agriculture, forestry, animal husbandry, and fishery | billion dollars |
| Extension of the agricultural industry chain | Total power of agricultural machinery/Total sown area of crops | kw/ha |
| | Total output value of agriculture, forestry, and fishery services/Total output value of agriculture, forestry, animal husbandry, and fishery | % |
| Expansion of agricultural multifunctionality | Agricultural fertilizer application rates/Total sown area of crop | ton/ha |
| | Grain sown area/Total sown area of crops | % |
| Industrial technology penetration | Number of rural broadband connections/Number of people in villages | pcs |
| | Mobile-phone penetration rate | pcs |
| Interest linkage mechanism | Per capita disposable income of farmers | dollars |
| | Urbanization rate | % |

Intra-agricultural integration refers to establishing organic links between various sub-industries within the agricultural industry and the effective integration of various resources, thereby promoting the linkage development of various sub-industries within the agricultural industry. Considering that the integration of resources within agriculture mainly relies on large-scale and mechanized production, this paper selects the indicator of the proportion of machine cultivation area to the total sown area of crops. At the same time, intra-agricultural integration aims to improve production efficiency, so this paper also selects the indicator of output per employee in agriculture, forestry, animal husbandry, and fishery to reflect the quality of intra-agricultural integration.

The extension of the agricultural industry chain refers to the extension of agricultural production as the center of the backward and forward industrial chains. The extension of the agricultural industry chain is mainly facilitated by industrial development, especially in the agricultural products processing industry. The ratio of the total power of agricultural machinery (including the food sector) to the total sown area and the share of the agriculture, forestry, animal husbandry, and fishery services in the total output value of agriculture, forestry, animal husbandry, and fishery industry are selected in this paper to reflect the quality of the extension of the agricultural industry chain.

The expansion of agricultural multifunctionality refers to the development of multiple functions of agriculture based on agriculture, implanted with the concepts of culture, leisure, and tourism, and promoting pro-environmental behavior. The realization of agricultural multifunctionality relies on sowing cash crops, such as flowers and sugar crops, while

considering the environmental protection of agricultural production. This paper selects the proportion of the sown area of food crops to the total sown area of crops and the amount of chemical fertilizer used per hectare to reflect the play of agricultural multifunctionality.

Industrial technology penetration refers to the transformation and upgrading of agriculture with the help of modern elements, with agriculture as the basic support. At the same time, considering that the penetration and application of technology mainly rely on the Internet, this paper selects the number of broadband access points per capita in rural areas and the mobile-phone penetration rate to reflect the penetration of industrial technology.

The interest linkage mechanism refers to how to drive farmers and enable them to gain benefits. The aim of developing rural industrial integration is to drive groups of farmers and allow them to share more in the benefits brought by the added value of the industry. At the same time, urbanization promotes the flow of resource factors between urban and rural areas, affecting the distribution of benefits between urban and rural populations. In this paper, two indicators, namely, the per capita disposable income of farmers and the urbanization rate, are chosen to measure the degree of perfecting the benefit linkage mechanism.

### 3.4. Data and Sample

This paper selects data from 31 provincial administrative regions in China from 2009 to 2019 as the research sample. Among them, data on the total output value of agriculture, forestry, animal husbandry, and fishery, the total output value of agriculture, forestry, animal husbandry, and fishery services, the number of employees in agriculture, forestry, animal husbandry, and fishery, the area under machine cultivation, the total sown area of crops, the per capita disposable income of farmers' households, rural electricity consumption, the number of the rural population, the amount of agricultural fertilizer used, the total power of agricultural machinery, the gross regional product, the mobile-phone penetration rate, the urbanization rate, the local financial expenditure, and the expenditure on agriculture, forestry, and water conservancy were obtained from the respective years of the *China Rural Statistical Yearbook*, the *China Statistical Yearbook*, the *China Population and Employment Statistical Yearbook*, and regional statistical yearbooks. Data on the number of students enrolled in higher agricultural colleges and universities at the undergraduate level were obtained from the *China Education Statistical Yearbook.* Table 2 reports the results of descriptive statistics for the main variables.

**Table 2.** Descriptive statistics of variables.

| Variables | Observations | Mean | SD |
|---|---|---|---|
| Road density | 341 | 8502.847 | 4903.305 |
| Rail density | 341 | 230.901 | 181.726 |
| Industrial structure | 341 | 0.164 | 0.082 |
| Economic size | 341 | 0.033 | 0.027 |
| Technical support | 341 | 0.032 | 0.023 |
| Human capital levels | 341 | 841.367 | 438.795 |
| Rural production infrastructure | 341 | 0.195 | 0.505 |

The dependent variable in this paper is the development quality of rural industrial integration. Transport infrastructure is the core explanatory variable in this paper. The academic community has mainly used transport costs [15], transport investment [19], and transport density [7] to measure the variable of transport infrastructure. Drawing on relevant studies [7,18] and taking into account the fact that road transport and railway transport account for a large proportion of rural industrial integration, this paper selects the road density and railway density of each region, i.e., the ratio of road mileage to the area of the region, to measure the transport infrastructure. The road density is selected as the ratio of the total road mileage to the region's area, including highways, primary roads, secondary roads, tertiary roads, and class 4 roads.

Control variables follow this. In this paper, factors that may have an impact on the quality of integrated rural industrial development, other than transport infrastructure, are set as control variables. These include the economic scale of each region, industrial structure, level of human capital, technical support, and the situation of rural production infrastructure. Firstly, each region's economic size is measured using each region's share of GDP of the total value of China's GDP. Secondly, the industrial structure is measured using the share of the output value of agriculture, forestry, animal husbandry, and fishery in the total output value. Third, on the level of human capital, the development of rural industrial integration requires the human capital support of laborers in the agricultural sector. This paper uses rural residents' education and cultural expenditure as a proxy variable to indicate the labor force situation in the region. Fourth, as regarding technical support, we draw on the studies of Sun and Wu [36,37], while the relevant technologies applied in the integration of rural industries can be developed and given technical support by higher agricultural colleges and universities. So, this paper uses the number of undergraduate students in higher agricultural colleges and universities as a proxy variable. Fifth, the situation of rural production infrastructure is measured by using the per capita electricity consumption of rural residents, which indicates the regional capital investment.

## 4. Results

### 4.1. Development Quality of China's Rural Industrial Integration

This paper uses the entropy weight method to establish an indicator system for evaluating the quality of integrated development of rural industries based on five dimensions: intra-agricultural integration, the extension of the agricultural industry chain, expansion of agricultural multifunctionality, industrial technology penetration, and interest linkage mechanism. The scores of rural industrial integration development in each provincial and municipal area in China each year are shown in Table 3. For space reasons, this paper only reports on the situation in each province and municipality in 2009, 2014, and 2019.

**Table 3.** Results of measuring the development quality of rural industrial integration in China.

| Region | 2009 | 2014 | 2019 | Region | 2009 | 2014 | 2019 |
|---|---|---|---|---|---|---|---|
| Beijing | 0.413 | 0.555 | 0.532 | Hubei | 0.147 | 0.249 | 0.430 |
| Tianjin | 0.252 | 0.324 | 0.468 | Hunan | 0.155 | 0.246 | 0.416 |
| Hebei | 0.159 | 0.260 | 0.395 | Guangdong | 0.238 | 0.353 | 0.480 |
| Shanxi | 0.164 | 0.247 | 0.330 | Guangxi | 0.118 | 0.198 | 0.391 |
| Inner Mongolia | 0.175 | 0.279 | 0.339 | Hainan | 0.155 | 0.272 | 0.449 |
| Liaoning | 0.236 | 0.352 | 0.370 | Chongqing | 0.119 | 0.218 | 0.393 |
| Jilin | 0.182 | 0.264 | 0.295 | Sichuan | 0.099 | 0.205 | 0.387 |
| Heilongjiang | 0.178 | 0.287 | 0.408 | Guizhou | 0.096 | 0.170 | 0.298 |
| Shanghai | 0.296 | 0.356 | 0.492 | Yunnan | 0.096 | 0.167 | 0.288 |
| Jiangsu | 0.253 | 0.427 | 0.737 | Tibet | 0.138 | 0.219 | 0.289 |
| Zhejiang | 0.292 | 0.434 | 0.692 | Shaanxi | 0.152 | 0.244 | 0.380 |
| Anhui | 0.132 | 0.217 | 0.432 | Gansu | 0.194 | 0.252 | 0.372 |
| Fujian | 0.212 | 0.362 | 0.641 | Qinghai | 0.134 | 0.206 | 0.343 |
| Jiangxi | 0.151 | 0.224 | 0.396 | Ningxia | 0.166 | 0.240 | 0.373 |
| Shandong | 0.208 | 0.315 | 0.479 | Xinjiang | 0.152 | 0.259 | 0.423 |
| Henan | 0.137 | 0.230 | 0.371 | — | — | — | — |

As shown in Table 3, firstly, the development quality of rural industrial integration in all 31 provincial-level administrative regions of China showed an upward trend from 2009 to 2019. Furthermore, rural industrial integration became a development trend, with all provinces and municipalities in China promoting the development of rural industrial integration. Secondly, the quality of development of rural industrial integration is generally higher in the provinces and cities located in the eastern region of China than in the western region of China. Thirdly, among the 31 provincial administrative regions, there is a huge

gap between the regions with a higher quality of integrated development of rural industries and those with lower quality of development. For example, in 2019, the level of rural industrial integration in Jiangsu, Zhejiang, and Fujian was 0.738, 0.692, and 0.641, while that in Guizhou, Tibet, and Yunnan was 0.298, 0.289, and 0.288, respectively. Fourthly, the pace of development of industrial integration is also highly uneven. The development of rural industrial integration in Jiangsu and Zhejiang is rapid, while the development of rural industrial integration in Beijing and Jilin has been slow.

*4.2. The Basic Regression Results*

4.2.1. Spatial Autocorrelation Analysis

As there may be a spatial correlation between regions, this paper applies the Moran test to measure the spatial correlation of rural industrial integration development quality in each provincial administrative region. Table 4 reports the specific results.

**Table 4.** Spatial autocorrelation analysis results.

| | $W^L$ | | $W^J$ | | | $W^L$ | | $W^J$ | |
|---|---|---|---|---|---|---|---|---|---|
| Year | Moran's | Z-Statistic | Moran's | Z-Statistic | Year | Moran's | Z-Statistic | Moran's | Z-Statistic |
| 2009 | 0.357 *** | 3.887 | 0.268 *** | 3.546 | 2015 | 0.243 *** | 3.718 | 0.167 *** | 3.259 |
| 2010 | 0.379 *** | 4.051 | 0.279 *** | 3.620 | 2016 | 0.239 *** | 3.981 | 0.172 *** | 3.666 |
| 2011 | 0.380 *** | 4.047 | 0.283 *** | 3.651 | 2017 | 0.230 *** | 3.952 | 0.166 *** | 3.666 |
| 2012 | 0.393 *** | 4.099 | 0.286 *** | 3.620 | 2018 | 0.220 *** | 4.024 | 0.163 *** | 3.875 |
| 2013 | 0.251 *** | 3.742 | 0.170 *** | 3.227 | 2019 | 0.217 *** | 4.023 | 0.171 *** | 4.071 |
| 2014 | 0.253 *** | 3.701 | 0.180 *** | 3.327 | — | — | — | — | — |

Note: *** indicate significant at the 1% levels of significance.

In Table 4, $W^L$ is the binary adjacency matrix using the Queen type, and $W^J$ is the inverse distance weight matrix using the latitude and longitude of each region. As shown in Table 4, the Moran index for rural industrial integration development quality in each of China's provincial administrative regions is positive. At the same time, all pass the test at the 1% significance level. This indicates a significant positive spatial correlation between the development quality of rural industrial integration in each provincial administrative region of China. Areas with a higher quality of development of rural industrial integration cluster with areas with a higher quality of development of rural industrial integration, while areas with a lower quality of development of rural industrial integration cluster with areas with a lower quality of development of rural industrial integration. Therefore, we should consider a spatial econometric model when exploring the impact of transport infrastructure on rural industrial integration. The model should add the spatial lag term *WY* of the dependent variable as the independent variable.

4.2.2. Benchmark Regression Result

As both the LM test and the Hausman test in the model selection process significantly rejected the original hypothesis, this paper chose the fixed-effects panel model for estimation. Moreover, the results of the LM-lag test indicated that this paper should apply a spatial autoregressive model. Therefore, this paper finally chooses a fixed-effects spatial panel autoregressive model to analyze the impact of transport infrastructure on rural industrial integration. To ensure the robustness of the regressions, we used road density and railway density as the core explanatory variables in the benchmark regression, respectively, to test the impact of transport infrastructure on the integration of rural industries. In addition, we used two spatial weight matrices, the Queen-type binary adjacency matrix in columns (1) and (3) of Table 5 and the inverse distance weight matrix in columns (2) and (4) of Table 5 to ensure that the regression results are robust. See Table 5 for specific regression results.

**Table 5.** Estimated results.

| Dependent Variable | Development Quality of Rural Industrial Integration | | | |
|---|---|---|---|---|
| Model | (1) | (2) | (3) | (4) |
| Road density | 0.002 *** | 0.002 *** | | |
| | (0.001) | (0.001) | | |
| Rail density | | | 0.122 *** | 0.113 *** |
| | | | (0.034) | (0.034) |
| Industrial structure | 95.811 | 82.982 | 104.765 | 90.554 *** |
| | (76.025) | (76.115) | (75.480) | (75.780) |
| Economic size | 760.686 ** | 641.158 * | 868.599 *** | 739.884 ** |
| | (330.345) | (332.125) | (328.876) | (331.561) |
| Technical support | −387.933 | −250.578 | −566.292 | −403.901 |
| | (429.949) | (427.758) | (430.711) | (429.340) |
| Human capital levels | 0.042 *** | 0.037 *** | 0.047 *** | 0.041 *** |
| | (0.012) | (0.012) | (0.012) | (0.012) |
| Rural production infrastructure | 20.598 *** | 14.234 ** | 22.099 *** | 15.152 ** |
| | (7.103) | (6.728) | (7.042) | (6.688) |
| Regional fixed effects | Yes | Yes | Yes | Yes |
| Year fixed effects | Yes | Yes | Yes | Yes |
| rho | 0.393 *** | 0.450 *** | 0.413 *** | 0.464 *** |
| | (0.063) | (0.071) | (0.062) | (0.071) |
| sigma2_e | 663.127 *** | 665.060 *** | 625.520 *** | 658.203 *** |
| | (51.578) | (51.649) | (50.854) | (51.169) |
| LogL | 757.217 | 757.104 | 759.211 | −353.995 |
| R-squared | 0.552 | 0.610 | 0.651 | 0.687 |

Note: *, **, and *** indicate significant at the 10%, 5% and 1% levels of significance respectively. Standard deviations are given in parentheses, same as in the table below.

The regression results reported in Table 5 show that the model parameters are estimated using the adjacency matrix and the distance matrix. Columns (1) and (2) of Table 5 use road density as the core explanatory variable, and columns (3) and (4) are replaced with railway density as the core explanatory variable. In the estimation results using the above two spatial weight matrices, the impact of transport infrastructure on rural industrial integration is consistent with our expected results, both reaching the 1% significance level. Moreover, the spatial spillover effect of rural industrial integration is also clearly present and positively driven, so hypotheses 1 and 2 proposed in this paper are verified. We can draw the following main conclusions based on the estimation results in Table 5.

The impact of transport infrastructure on the integration of rural industries. Firstly, the impact of local road density on the quality of local rural industrial integration development is positive, and all of them pass the 1% significance level test. This result indicates that the greater the density of local roads, the more favorable the development of rural industrial integration. Secondly, the effect of local railway density on the quality of local rural industrial integration development is positive, and all pass the 1% significance level test. This result indicates that the greater the density of railways in the region, the more favorable it is to develop rural industrial integration.

We analyzed the spatial spillover effect of transport infrastructure on rural industrial integration. Firstly, the estimated results in Table 5 show that the coefficients of the spatial autoregressive terms are all significantly positive and pass the 1% significance level test. This result indicates that the development of rural industrial integration in geographically proximate areas will improve the quality of development of local rural industrial integration through transport infrastructure. The development of rural industrial integration has a positive external effect. Secondly, the spatial spillover effect brought about by transport infrastructure as a medium for the spillover effect of rural industrial integration is positive, all passing the 1% significance level test. Compared with road traffic, the coefficient of the spatial lag term of the development quality of rural industrial integration is higher when the model is estimated using railway traffic as the core variable, which indicates that the

impact of railway traffic is higher than that of road traffic when the spatial spillover effect of rural industrial integration is brought into play.

### 4.3. Heterogeneity Analysis

4.3.1. Analysis of the Heterogeneity of the Time Dimension

In 2014, China's Central Rural Work Conference proposed "introducing modern industrial organization methods, such as industrial chains and value chains, into agriculture and promoting the integrated development of the primary, secondary, and tertiary industries in rural areas", which was the first time that rural industrial integration was officially mentioned at the national level. Therefore, this paper takes 2014 as an important point and divides the sample data into two parts, 2009–2013 and 2014–2019, to analyze the impact of transport infrastructure on the integration of rural industries in different time stages. See Table 6 for specific regression results.

**Table 6.** Heterogeneity estimation results in the time dimension.

| Dependent Variable | Development Quality of Rural Industrial Integration | | | |
|---|---|---|---|---|
| Year | 2009–2013 | | 2014–2019 | |
| Model | (1) | (2) | (3) | (4) |
| Road density | 0.001 * (0.001) | | 0.003 ** (0.001) | |
| Rail density | | 0.074 (0.063) | | 0.112 *** (0.036) |
| Control variables | Yes | Yes | Yes | Yes |
| Regional fixed effects | Yes | Yes | Yes | Yes |
| Year fixed effects | Yes | Yes | Yes | Yes |
| rho | 0.172 (0.106) | 0.162 (0.109) | 0.470 *** (0.073) | 0.493 *** (0.071) |
| LogL | −630.000 | −631.013 | −785.260 | −782.822 |
| R-squared | 0.003 | 0.022 | 0.432 | 0.544 |

Note: *, **, and *** indicate significant at the 10%, 5% and 1% levels of significance respectively.

The spatial weight matrix used in this section is a Queen-type binary adjacency matrix. With road transport as the core explanatory variable, Table 6 reports the results: first, the spatial lag term for the development quality of rural industrial integration with the help of road transport facilities is more significant in the period 2014–2019 compared to 2009–2013, passing the 1% significance level test. This result indicates that after the Chinese government officially proposed rural industrial integration at the national level in 2014, regions paid more attention to its development, and inter-regional interaction increased. The spatial spillover effect of promoting rural industrial integration through road transport facilities came to the fore. Secondly, from 2014 to 2019, rural industrial integration was given more attention, and the flow of factors involved was active. The positive impact of road transport facilities on the development quality of rural industrial integration was more significant than in 2009–2013, passing the test at the 5% significance level.

With railway transportation as the core explanatory variable, the estimation results from Table 6 show that: firstly, from 2014 to 2019, the impact of railway transportation facilities on rural industrial integration becomes significant as the development of rural industrial integration is emphasized, passing the test at a 1% significance level, while the spatial spillover effect of rural industrial integration with the help of railway transportation is significant, passing the 1% significance level test. Secondly, the variable of railway density did not pass the significance level test during 2009–2013. This result may explain that the integration of rural industries did not receive much attention during this period, so there were fewer links between regions regarding the integration of rural industries. The spatial lag term of rural industrial integration development quality was insignificant.

The development of the integration of rural industries in each region was more likely to be carried out within the region through road transportation.

4.3.2. Analysis of the Heterogeneity of the Spatial Dimension

As China is a vast country with significant differences between regions, this section draws on existing research [38]. It examines the impact of transport infrastructure on the development of rural industrial integration in each region, according to the classification of eastern and midwest regions. See Table 7 for specific regression results.

**Table 7.** Heterogeneity estimation results in the spatial dimension.

| Dependent Variable | Development Quality of Rural Industrial Integration | | | |
|---|---|---|---|---|
| Region | East | | Midwest | |
| Model | (1) | (2) | (3) | (4) |
| Road density | 0.006 *** (0.002) | | 0.002 * (0.001) | |
| Rail density | | 0.008 (0.103) | | 0.153 *** (0.039) |
| Control variables | Yes | Yes | Yes | Yes |
| Regional fixed effects | Yes | Yes | Yes | Yes |
| Year fixed effects | Yes | Yes | Yes | Yes |
| rho | 0.306 *** (0.082) | 0.316 *** (0.083) | 0.161 * (0.096) | 0.225 ** (0.095) |
| LogL | −697.925 | −701.249 | −914.423 | −908.824 |
| R-squared | 0.064 | 0.061 | 0.712 | 0.730 |

Note: *, **, and *** indicate significant at the 10%, 5% and 1% levels of significance respectively.

From the estimation results in Table 7: firstly, road density in the eastern region positively affects the quality of development of local rural industrial integration and passes the 1% significance level test. The estimated coefficients and significance of road density in the eastern region are higher than those in the midwest regions, indicating that the impact of road transportation facilities on local rural industrial integration is more significant in the eastern region. Secondly, the impact of railway density on the integration of rural industries in the midwest regions is more significant than that in the eastern regions, with a positive effect, passing the 1% significance level test. This result indicates that the midwest regions rely more on railway transportation to develop the integration of rural industries or the deployment of resources. Thirdly, regarding the spatial lag of rural industrial integration development quality, the eastern and midwest regions are significant and pass the 10% significance level test. However, the spatial lag of the development quality of rural industrial integration in the eastern region passes the 1% significance level, which is more significant than that in the midwest regions, probably because the eastern provinces and cities are mainly developed regions—they have more frequent contact with the surrounding areas in their development. In contrast, the midwest provinces and cities are less developed regions and do not have very close contact with their neighbors.

## 5. Discussion

### 5.1. China's Rural Industrial Integration Development Pattern

To promote the high-quality development of rural industries, the Chinese government has proposed the initiative of rural industrial integration. This paper uses the entropy method to construct a rural industrial integration evaluation index system to measure the development quality of rural industrial integration in China. Different from the established studies [35,39], this study adds farmers' income and urbanization rate to the evaluation indexes to improve the index system based on the *Guidelines on Promoting the Integration of Rural Primary, Secondary, and Tertiary Industries* issued by the General Office of the State Council of China in 2015, and according to the deployment of close interest linkage and

urban–industry integration therein. Thirty-one provincial administrative regions in China actively promoted the development of rural industrial integration from 2009 to 2019, and the quality of rural industrial integration development in all regions shows an upward trend [34]. In particular, the quality of development of rural industrial integration in China's provincial and municipal regions rose more significantly during the period 2014–2019, and the growth rate of rural industrial integration scores in many regions during this period exceeded that of the period 2009–2013 [40]. Linking the development of rural industrial integration with the policies promulgated by the Chinese government, it can be seen that since the central government of China officially proposed the development of rural industrial integration nationwide in 2014, and all regions of China began to pay attention to and actively promote the development of rural industrial integration.

However, while recognizing the steady development of industrial integration in China's rural areas, we should also look at the disparities in the development of industrial integration in China's rural areas by region. Firstly, the development quality of rural industrial integration in China shows a pattern of "high in the east and low in the west". In terms of the geographical location of each province and city, the provinces and cities with high-quality rural industrial integration development, such as Jiangsu, Zhejiang, Fujian, and Beijing, are all located in the eastern part of China. In contrast, Tibet, Yunnan, and Guizhou, where the development of rural industrial integration has been slow, are all located in China's midwest regions [39], indicating significant regional differences in rural industrial integration across China. Secondly, the pace of development of rural industrial integration is also highly uneven across provinces and cities in China [40]. The Yangtze River Delta region, such as Jiangsu and Zhejiang, has been developing rural industrial integration at a faster pace, while the development of rural industrial integration in Jilin and Liaoning has always been slower. The development of rural industrial integration is uneven among the regions of Chinese provinces and cities. In order to promote the high-quality development of China's rural industries, we can efficiently use China's national territory by planning transportation facilities and rationalizing the construction of economic zones.

## 5.2. The Impact of Transport Infrastructure on the Quality of Rural Industrial Integration

The results of this study confirm that both road and rail transport positively affect the development of local rural industrial integration. There is a basic consensus in relevant studies that transport infrastructure facilitates the development of regional economies and industries [41,42], and for the development of agriculture and rural industries, the driving role of transportation infrastructure remains [43]. Existing studies have provided evidence from Africa [44,45], Australia [46], and the Americas [47]. Using data from China, this study confirms the role of transportation infrastructure as a driver of rural industrial integration. The explanation is that better transport infrastructure facilitates the flow of factors and resource allocation [48,49], and, at the same time, it can break the market segmentation [50]. Efficient freight transport, and convenient passenger transport, will promote the development of rural industrial integration. The transport involved in rural industries integration, such as bulk grain crops, requires railway transport. At the same time, intra-regional connectivity will be more by road, while inter-regional connectivity between neighboring regions, or long-distance transport, will rely more on railway transport. The more developed railroad transportation is, the lower the transportation cost of production factors with other regions, and the more frequent the flow of resources will be [51], which will favor the development of local rural industrial integration. The focus will be on promoting the development of rural industrial integration through transportation facilities. Road [52] and railroad [53] facilities can promote development. Therefore, the most suitable transportation mode should be chosen according to different situations and industries to develop the integration of rural industries and deploy the resources and factors required. At the same time, the efficiency of transportation and factor flow can be improved, and its cost reduced through the connection and combination of different transportation modes to drive the development of the integration of rural industries. While improving the degree of

road and railroad construction, the transportation support measures are updated with the help of information and weather forecasting technologies [46]. High-quality agricultural development is promoted through transportation planning.

Strengthening the transport infrastructure construction is conducive to improving the quality of rural industrial integration. At the same time, with the help of transport infrastructure, the development of local rural industrial integration will drive the development of rural industrial integration in neighboring areas. Existing studies have demonstrated spatial spillover effects of regional industrial development through transportation facilities [28,54]. This study was conducted around rural industrial integration, which also confirmed the existence of spatial spillover effects. By linking different areas, transport facilities facilitate the concentration of factors in the most efficient areas [55,56], allowing for the diffusion effect of the integration of rural industries. The transport infrastructure serves as a bridge to connect the flow of production factors between regions [57], enabling the positive external effect of rural industrial integration to be brought into play. Railway traffic is more conducive to interacting and exchanging rural industrial integration between different regions [51]. Connecting various regions through transportation facilities while focusing on transportation diversification brings into play the positive external effects of rural industrial integration. Given the wide gap in the quality of development of rural industrial integration in different regions, the government can improve regional transport infrastructure, build road and railway networks [57], and open up the "last mile" [46] to promote the deployment of resources between regions and improve the efficiency of factor production. The areas with better development of rural industrial integration drive the surrounding areas with slower development of rural industrial integration, narrow the development gap of rural industrial integration between regions, and realize the joint improvement of the quality of rural industrial integration development between regions.

The heterogeneity analysis part of this study shows that transportation facilities' impact on rural industrial integration varies across regions and development stages. The differences in the characteristics of different regions have been noticed by scholars and have been the focus of heterogeneity studies [26]. However, we note that different stages of development of rural industrial integration also present different characteristics. Therefore, we pay special attention to the results of different stages in the section of the heterogeneity study and discuss the impact of transportation facilities on rural industrial integration at different development stages separately. Road transport has a more significant impact on developing rural industrial integration in eastern China. In contrast, railway transport has a more significant impact on developing rural industrial integration in midwest China. The possible reasons are that the eastern regions are mainly developed areas with better road transport facilities [26]. At the same time, the regional area of the eastern regions is smaller than that of the midwest regions. Factor flows rely on road transport, so the impact of road density on integrating rural industries in the eastern regions is significant. The midwest regions are vaster and, at the same time, less developed, and the development of road transportation is not well developed. In order to reduce the costs, agriculture relies more on rail transportation for resource deployment [57]. From the stage analysis, we can see that when rural industrial integration was not widely noticed, each region developed more independently, and inter-regional communication was infrequent. So rural industrial integration was mainly carried out within the local area using road transportation. Whereas during 2014–2019, the Chinese government attached great importance to the development of rural industrial integration, and the development of rural industrial integration between regions was rapid, and communication began to be frequent. The impact of railway transportation rose significantly [58]. Given the differences in the development of rural industrial integration in different periods and regions, the use of transport facilities is tailored to the time and place [59], and road transport is used to promote the development of local rural industrial integration and the deployment of resource factors. In contrast, railway transport strengthens communication between neighboring regions [60]. Meanwhile, it is appropriate for the eastern region to prefer road transportation [26], and it is

feasible for the midwest regions to prefer rail transportation. Rational planning of China's national resources through the layout of transport facilities and promoting high-quality development of China's rural industries. However, this study uses provincial data and can only provide insights into practice at the level of provincial areas and cannot be refined at the municipal level.

## 6. Conclusions

This study to discuss whether transport infrastructure has promoted the development of rural industrial integration empirically examines the impact of transport infrastructure on rural industrial integration through the entropy weight method and spatial panel autoregressive model with panel data from 31 provincial administrative regions in China from 2009 to 2019. The study shows that, firstly, based on the measurement of the quality of rural industrial integration development in each region. The quality of rural industrial integration development in each region of China from 2009 to 2019 shows an upward trend, but the development is uneven. It shows a "high in the east and low in the west" pattern. Second, the analysis is based on spatial econometrics. Transportation infrastructure promotes the development of rural industrial integration. With the help of transport infrastructure, the improved quality of development of rural industrial integration in each region will drive the development of rural industrial integration in the surrounding areas. After 2014, the Chinese government began to pay attention to developing rural industrial integration. After this period, the quality of rural industrial integration in various regions has improved, and the spatial spillover effect is pronounced. The development of rural industrial integration in eastern China is more significantly influenced by local road transport facilities, while local railway transport facilities more significantly influence the development of rural industrial integration in midwest China.

Based on the findings of this part of the study, we confirm the positive impact of transport infrastructure on the integration of rural industries. Therefore, to promote the integration of rural industries and drive rural areas' industrial high-quality development, the government can consider supporting them and promoting their development by improving transport planning. It is also conducive to the realization of scientific land use. This study proposes the following policy recommendations: (1) focus on the role of transportation planning in promoting high-quality agricultural development; (2) emphasis to be placed on diversified transportation facility applications to leverage the spillover effect of rural industrial integration; and (3) the precise use of transportation supports high-quality agricultural development, depending on time and region.

This study also has some potential limitations. First, in terms of data and sample, this study uses data at the provincial level in China and has a period of only 11 years. Further research would have been more detailed and richer if the data at the municipal level had been chosen, along with a larger period. Second, according to the production function, capital, technology, and labor are all important factors that affect production efficiency. However, this paper focuses on the influence of transportation infrastructure, and capital, technology, and labor are only used as control variables. It is a meaningful question as to what results will emerge when capital, technology, and labor are added as mediating variables in studying transportation facilities for high-quality agricultural development. It is also the next research direction. Third, this study is centered on the high-quality development of agriculture and land planning in China, and the findings need further verification for other countries and regions.

**Author Contributions:** D.W. contributed to conceptualization; H.Z., collected the data and contributed econometrics analysis and writing. All authors have read and agreed to the published version of the manuscript.

**Funding:** This paper was supported by the National Social Science Foundation of China (01095818002).

**Institutional Review Board Statement:** Not applicable.

**Informed Consent Statement:** Not applicable.

**Data Availability Statement:** The data will be made available on request.

**Conflicts of Interest:** The authors declare no conflict of interest.

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
