# Peer review of "The Impact of Transport Infrastructure on Rural Industrial Integration: Spatial Spillover Effects and Spatio-Temporal Heterogeneity"

_land, doi:10.3390/land11071116_

Round 1

Reviewer 1 Report

The research is well structured and presented. I suggest further research on the same problem.

Author Response

Point 1:The research is well structured and presented. I suggest further research on the same problem. 

Response 1:Thank you for your letter and comments concerning our manuscript. Thank you very much for your recognition of our work. Your approval also makes us feel more confident about the subsequent research. I wish you all the best.

Reviewer 2 Report

The manuscript titled " The Impact of Transport Infrastructure on Rural Industrial Integration: Spatial Spillover Effects and Spatio-Temporal Heterogeneity " intends to answer the follow questions: Does the improvement of transport facilities impact the development of rural industrial integration? What impact will it have on the development of rural industrial integration? The manuscript select data from 31 provincial administrative regions in China from 2009 to 2019. It uses the entropy weight method and spatial panel autoregressive model to empirically test the impact of transport infrastructure on the integration of rural industries.

The research is original; it could be characterized as novel and in my opinion important to the field, it also has an almost appropriate structure and the language has been used well. In the meanwhile, the manuscript has a good extent (about 8,150 words) and it is comprehensive. The tables (7) and figures (3) make the paper to reflect well to the reader. For this reason, paper has a "diversity look", not only tables, not only numbers, not only words. It is advised to revise figures, compare them, or use appendix if you agree.

The title, I think, is all right. The abstract did not reflect well the findings of this study, and it has not the appropriate length. Please revise the abstract of the manuscript and do not forget abstract need to encourage readers to download the paper. The Abstract needs further work. It is not clear. Abstracts should indicate the research problem/purpose of the research, provide some indication of the design/methodology/approach taken, the findings of the research and its originality/value in terms of its contribution to the international literature. The abstract has a long length (about 276 words). Please, revise the abstract, it must be up to 200 words long, for this reason I would be good to reduce [see: Instructions for Authors / Manuscript Submission Overview / Accepted File Formats - (https://www.mdpi.com/journal/land/instructions#submission or https://www.mdpi.com/files/word-templates/land-template.dot)].

Please, revise the manuscript and make the appropriate currency exchange using dollars ($) or euros (€) or both also you can keep yuan (see Table 1 – you don’t have to change numbers). This is because the results of the research must be directly comparable to other similar surveys that have already been carried out around the world and other such surveys will certainly be carried out, and do not forget, the journal “Land” is international. Moreover, in Table 1, second line say something about billion (€, $ or yuan).

The introduction is effective, clear, and well organized; it really introduced and put into perspective what research is negotiating but is very big. Please revise the Introduction of the manuscript and include references which are already exists in bibliography (as you did). For the Methodology chapter, the research conduct has been tested in several areas of the world, with similar results and will probably be tested in others. Appropriate references to the methodology included in the already published bibliography. It is advised to revise (a little) the Discussion and Conclusion. Both sections should be consistent in terms of Proposal, Problem statement, Results, and of course, future work. Your conclusion section is big and does not justice to your work. Make it your key contributions, arguments, and findings clearer. You must refer to the literature and previous studies in your discussion section.

More discussion is needed, comparing the results of this work related to attributes with those of other studies. I believe that the conclusions section or discussion should also include the main limitations of this study and incorporate possible policy implications (as you did). I think, something more should be said about practical implications.

Please revise the references of the manuscript and include references which are already exists in bibliography. I would be much more satisfied if the number of references was slightly higher (about 20 - 25 references) and I would appreciate it if also included data from the entire world (Asia, America, Europe and Australia e.tc.), not only from Asia, because Land is also an international Journal. In this way it is documented that a project which is tested in a place with its own characteristics can be implemented in other places around the world. References must have an appropriate style, for this reason I would be good to change [see: Instructions for Authors / Manuscript Preparation / Back Matter / References: - (https://www.mdpi.com/journal/land/instructions or https://www.mdpi.com/authors/references)]. Do not forget, DOI numbers (Digital Object Identifier) are not mandatory but highly encouraged and make the review easier.

For example, for reference 21 you write “Faber,B. Trade Integration, Market Size, and Industrialization: Evidence from China's National Trunk Highway System, Review of Economic Studies,2014, 1046—1070”. I think must be revised as “Faber, B. Trade Integration, Market Size, and Industrialization: Evidence from China’s National Trunk Highway System. Rev. Econ. Stud. 2014, 81, 1046–1070”.

Author Response

Dear Reviewers:

I am very grateful for your comments on the manuscript. Those comments are valuable and helpful for revising and improving our paper and the important guiding significance to our research. We have studied the comments carefully and made a correction that we hope meets with approval. Revised portions are marked in red on the paper. According to your advice, we amended the relevant part of the manuscript. Some of your questions were answered below.

Reviewer’s comments: For this reason, paper has a "diversity look", not only tables, not only numbers, not only words. It is advised to revise figures, compare them, or use appendix if you agree.

Response: Thanks for your comments. Based on your suggestion and consideration of the style of the paper, we agreed to place the figures in the appendix. In the resubmitted manuscript, we have placed the figures (Schematic diagram of rural industrial integration development in 31 provincial administrative regions of China) in the appendix. We have also made a comparison, briefly explained in text form in the appendix.

Reviewer’s comments: The abstract did not reflect well the findings of this study, and it has not the appropriate length. Please revise the abstract of the manuscript and do not forget abstract need to encourage readers to download the paper. The Abstract needs further work. It is not clear. Abstracts should indicate the research problem/purpose of the research, provide some indication of the design/methodology/approach taken, the findings of the research and its originality/value in terms of its contribution to the international literature. The abstract has a long length (about 276 words). Please, revise the abstract, it must be up to 200 words long, for this reason I would be good to reduce.

Response: Thanks for your comments. We are very sorry for our negligence regarding abstract requirements. We have re-written this part in the revised manuscript according to the Reviewer’s suggestion. In the new abstract, we first describe the background and purpose of our study. After that, we describe the methods we used to conduct the research and what we did. Third, we present the findings of our study; and finally, we present the implications of our study and possible contributions to related research in other parts of the world. At the same time, we paid great attention to the word count requirements in the abstract section while revising the abstract. The new abstract totals 199 words.

Reviewer’s comments: Please, revise the manuscript and make the appropriate currency exchange using dollars ($) or euros (€) or both also you can keep yuan (see Table 1 – you don’t have to change numbers). This is because the results of the research must be directly comparable to other similar surveys that have already been carried out around the world and other such surveys will certainly be carried out, and do not forget, the journal “Land” is international. Moreover, in Table 1, second line say something about billion (€, $ or yuan).

Response: Thanks for your comments. We have revised this section. First, Considering the journal “Land” is international, we have changed the currency to dollars in this section of the revision. We think this can be more acceptable to the readers. After that, we changed billions to billions dollars in the second row of Table 1. We are sorry for this oversight.

Reviewer’s comments: Please revise the Introduction of the manuscript and include references which are already exists in bibliography (as you did). For the Methodology chapter, the research conduct has been tested in several areas of the world, with similar results and will probably be tested in others. Appropriate references to the methodology included in the already published bibliography. It is advised to revise (a little) the Discussion and Conclusion. Both sections should be consistent in terms of Proposal, Problem statement, Results, and of course, future work. Your conclusion section is big and does not justice to your work. Make it your key contributions, arguments, and findings clearer. You must refer to the literature and previous studies in your discussion section.

Response: Thanks for your comments. We have revised the introduction, research methods, discussion, and conclusion sections. First, we have revised the introduction section. The new introduction section is more focused on industry and rural industrial integration, which can make our discussion appear more specific. At the same time, we cut unnecessary language from the introduction to make our study appear less large. This makes our introduction section more concise and clear. Second, we revised the research methods section. We included literature in the research methods section to support the choice of our research methods and to highlight the rationality of our research methods. Third, we revised the conclusion section. The conclusion section was made consistent with the content of the discussion section. The new conclusion section mainly presents our contributions and findings, as well as a summary of the practical insights of the discussion section and limitations. Also, the new conclusion is more specific. Finally, we modified the discussion section. Reference to relevant studies with joining literature.

Reviewer’s comments: More discussion is needed, comparing the results of this work related to attributes with those of other studies. I believe that the conclusions section or discussion should also include the main limitations of this study and incorporate possible policy implications (as you did). I think, something more should be said about practical implications.

Response: Thanks for your comments. We have modified the discussion section more to make our discussion section more complete. First, we included literature from relevant studies and compared our analysis with other studies to highlight the differences in our study. Second, we mention the limitations of our research and the limitations of the policy implications associated with it. Finally, we add in the discussion section possible practical insights from our study that we hope will positively impact practical developments.

Reviewer’s comments:Please revise the references of the manuscript and include references which are already exists in bibliography.

Response: Thanks for your comments. We have added references. Excluding some older literature, including additions, the references now total 60. The literature also includes relevant studies from Australia, the United States, Africa, and Europe, which makes our manuscript more informative. Finally, we revised the references format, including adding missing information, changing the journal names to abbreviations, and adding DOI.

Thank you very much for your suggestion. Your suggestions are beneficial for us in revising our manuscript. I wish you all the best.

Reviewer 3 Report

Journal: Land (ISSN 2073-445X)
Manuscript ID: land-1810807
Type: Article
Number of Pages: 19

Title: The Impact of Transport Infrastructure on Rural Industrial Integration: Spatial Spillover Effects and Spatio-Temporal Heterogeneity

Dear Authors,

It has been for me a great honour, as well as a pleasantly challenging activity, to review the article entitled ”The Impact of Transport Infrastructure on Rural Industrial Integration: Spatial Spillover Effects and Spatio-Temporal Heterogeneity.

Overall, the article is interesting and easy to read. However, I suggest that the Authors introduce a few corrections (given below).

In my opinion, the Introduction chapter well introduces potential readers to the topics discussed by the Authors. It is based on well-chosen literature. The aim and the marginal contributions of the paper are clearly stated.

Minor remarks:

Line 31 (“In 1963, Rosenberg put forward the concept of industrial convergence”) - source (literature reference) is missing. Besides, in the first paragraph the Authors provide a lot of data, but do not cite any sources.

Line 77 (“We summarized”) - it would be more correct to write in the passive voice. However, I will not insist on it, I leave it for the Authors' consideration.

Research Hypothesis

This chapter is readable, based on well-chosen literature and well-illustrated by Figure 1.

Research Methodology – is written in a clear and understandable way. It is logically divided into the following four subchapters. Table 1 and 2 are a good illustration. However, a greater support in methodical literature would also be valuable, which would justify the application of the chosen research method.

Results

This chapter presents the results of the research obtained in an understandable way. It is also logically divided into the following subchapters and subsections. It is well illustrated by 5 tables and 2 figures.

Discussion

The discussion is conducted in a correct and interesting manner. However, it could be extended and to a greater extent the research results obtained by the Authors could be confronted with other publications on this subject. In subsection 5.1. a lot of information is given, but again no literature references (no source for this information).

5.Conclusions

In my opinion, the conclusions are correct ale clearly stated. It is worth emphasizing that the article is also of an utilitarian nature, it contains policy recommendations that can be used in practice. I also consider it beneficial that the Authors indicated the limitations encountered and the direction of future research.

I don't feel competent to comment on linguistic correctness as English is not my mother tongue. I can only say that the text reads well and the article has a good chance of attracting potential readers. I wish the Authors good luck.

Author Response

Dear Reviewers:

Thank you for your letter and comments concerning our manuscript entitled “The Impact of Transport Infrastructure on Rural Industrial Integration: Spatial Spillover Effects and Spatio-Temporal Heterogeneity” (ID: land-1810807). Those comments are valuable and very helpful for revising and improving our paper and the important guiding significance to our research. We have studied the comments carefully and made a correction that we hope meets with approval. Revised portion are marked in red in the paper. The following part is the point-by-point responses to the Reviewers:

Reviewer’s comments: Line 31 (“In 1963, Rosenberg put forward the concept of industrial convergence”) - source (literature reference) is missing. Besides, in the first paragraph the Authors provide a lot of data, but do not cite any sources.

Response: Thanks for your comments. We are very sorry for our negligence of the source. In the revised manuscript, we have added the corresponding literature, along with a description of the sources of the data and documents.

Reviewer’s comments: Line 77 (“We summarized”) - it would be more correct to write in the passive voice. However, I will not insist on it, I leave it for the Authors' consideration.

Response: Thanks for your comments. We think your suggestion can make the manuscript of our paper more correct. In the revised manuscript, We changed this sentence to the passive voice.

Reviewer’s comments: However, a greater support in methodical literature would also be valuable, which would justify the application of the chosen research method.

Response: Thanks for your comments. We have added literature in the methodology section to support the soundness of our study.

Reviewer’s comments: The discussion is conducted in a correct and interesting manner. However, it could be extended and to a greater extent the research results obtained by the Authors could be confronted with other publications on this subject. In subsection 5.1. a lot of information is given, but again no literature references (no source for this information).

Response: Thanks for your comments. We have made changes in the discussion section. First, we refer to other publications for relevant studies, Cite these studies, and compare our analysis with them. Second, we supplemented the literature in Section 5.1 to corroborate our views with relevant studies.

Thank you very much for your suggestion. Your suggestions are beneficial for us in revising our manuscript. I wish you all the best.